# Chromosome-Level Assemblies for the Pine Pitch Canker Pathogen *Fusarium circinatum*

**DOI:** 10.3390/pathogens13010070

**Published:** 2024-01-12

**Authors:** Lieschen De Vos, Magriet A. van der Nest, Quentin C. Santana, Stephanie van Wyk, Kyle S. Leeuwendaal, Brenda D. Wingfield, Emma T. Steenkamp

**Affiliations:** 1Department of Biochemistry, Genetics and Microbiology (BGM), Forestry and Agricultural Biotechnology Institute (FABI), University of Pretoria (UP), Pretoria 0002, South Africa; lieschen.bahlmann@fabi.up.ac.za (L.D.V.); kyle.leeuwendaal@fabi.up.ac.za (K.S.L.); brenda.wingfield@fabi.up.ac.za (B.D.W.); 2Hans Merensky Chair in Avocado Research, Department of Biochemistry, Genetics and Microbiology, Forestry and Agricultural Biotechnology Institute FABI, University of Pretoria, Pretoria 0002, South Africa; magriet.vandernest@fabi.up.ac.za; 3Biotechnology Platform, Agricultural Research Council, 100 Old Soutpan Road, Onderstepoort, Pretoria 0010, South Africa; santanaq@arc.agric.za; 4Collaborating Centre for Optimising Antimalarial Therapy (CCOAT), Mitigating Antimalarial Resistance Consortium in South-East Africa (MARC SEA), Department of Medicine, Division of Clinical Pharmacology, University of Cape Town, Cape Town 7925, South Africa; stephanie.vanwyk@gmail.com

**Keywords:** dispensable chromosome, centromere, telomere, intrachromosomal translocation

## Abstract

The pine pitch canker pathogen, *Fusarium circinatum*, is globally regarded as one of the most important threats to commercial pine-based forestry. Although genome sequences of this fungus are available, these remain highly fragmented or structurally ill-defined. Our overall goal was to provide high-quality assemblies for two notable strains of *F. circinatum*, and to characterize these in terms of coding content, repetitiveness and the position of telomeres and centromeres. For this purpose, we used Oxford Nanopore Technologies MinION long-read sequences, as well as Illumina short sequence reads. By leveraging the genomic synteny inherent to *F. circinatum* and its close relatives, these sequence reads were assembled to chromosome level, where contiguous sequences mostly spanned from telomere to telomere. Comparative analyses unveiled remarkable variability in the twelfth and smallest chromosome, which is known to be dispensable. It presented a striking length polymorphism, with one strain lacking substantial portions from the chromosome’s distal and proximal regions. These regions, characterized by a lower gene density, G+C content and an increased prevalence of repetitive elements, contrast starkly with the syntenic segments of the chromosome, as well as with the core chromosomes. We propose that these unusual regions might have arisen or expanded due to the presence of transposable elements. A comparison of the overall chromosome structure revealed that centromeric elements often underpin intrachromosomal differences between *F. circinatum* strains, especially at chromosomal breakpoints. This suggests a potential role for centromeres in shaping the chromosomal architecture of *F. circinatum* and its relatives. The publicly available genome data generated here, together with the detailed metadata provided, represent essential resources for future studies of this important plant pathogen.

## 1. Introduction

The era of fungal genomics began in 1996 with the sequencing of the baker’s yeast (*Saccharomyces cerevisiae*) genome [1]. In 2003, the genome of the first filamentous fungus, *Neurospora crassa*, was sequenced [2]. These two milestones in genome research provided the impetus for subsequent fungal work and, by 2010, the genome sequences for more than 100 fungi were available in public databases [3]. Towards the end of 2023, one of the main repositories for fungal genomes, MycoCosm (https://mycocosm.jgi.doe.gov, accessed on 1 November 2023), contained data for more than 2500 species, spanning the fungal tree of life, with thousands more genomes currently being sequenced. The collective availability of these resources has revolutionized our knowledge of the ecology, evolution and overall biology of fungi [4,5,6,7,8,9]. Indeed, genome sequencing has become a routine part of modern fungal research.

The socioeconomically important genus *Fusarium* (phylum, *Ascomycota*; family, *Nectriaceae*; and order, *Hypocreales*) provides an illustrative example of how technological advances and decreasing costs have driven large-scale genome initiatives [4,10,11,12]. Apart from genome sequences being publicly available for numerous *Fusarium* species, multiple strains of many agriculturally/medically important species have also been sequenced. One such species is the pine pitch canker pathogen, *Fusarium circinatum*; the whole genome sequences of 17 strains are currently available in the database of the National Center for Biotechnology Information (https://www.ncbi.nlm.nih.gov/; accessed on 1 November 2023). Together with other socioeconomically important species, this pathogen forms part of the so-called *Fusarium fujikuroi* species complex (FFSC) [13]. It can infect more than 60 *Pinus* species at all growth stages and is globally regarded as one of the most important risks to pine-based forestry enterprises [14]. The pathogen typically causes chlorosis and the development of large resinous cankers at infection points on the trunks and branches of trees, often leading to dieback. In the case of younger plants, especially in seedlings in commercial nurseries, infection by *F. circinatum* causes the wilting and chlorosis of needles due to severe root and root collar diseases, culminating in plant mortality. The need for effective control strategies has thus sparked considerable interest in the use of genomic tools for studying the genetics, evolution and general biology of this important pathogen.

*Fusarium circinatum* was the first eukaryote to have its whole genome sequenced on the African continent [15]. As is the case for most publicly available fungal genomes, this was conducted using second-generation sequencing technologies. The original assembly was compiled for strain FSP34 using the 454 GS FLX system [15] and later augmented with SOLiD^TM^ mate-pair data, but the resulting assembly remained highly fragmented and incomplete [16]. The same is true for the second strain (KS17) that was sequenced using SOLiD mate-pair data [17]. Despite their limited quality, however, these data were invaluable for improving our knowledge of the pitch canker pathogen, especially in terms of its population dynamics and pathogenesis. Furthermore, by making use of the genetic linkage map for strain FSP34 and the macrosyntenic nature of FFSC genomes [18], contigs comprising the initial genome data for FSP34 and KS17 could be ordered into pseudomolecules corresponding to the twelve chromosomes of *F. circinatum* [16].

The advent of third-generation technologies enabled the real-time reading of nucleotide sequences at the single molecule level [19], allowing for the production of long sequence reads (>2.27 Mb) [20]. A prominent example of such a long-read system is the portable MinION sequencer from Oxford Nanopore Technologies (ONT), which uses nanopores to sequence a single DNA molecule per pore, bypassing the sequencing-by-synthesis method of traditional sequencers [21]. The high error rate inherent to this system is then accounted for by using its long-read output in conjunction with short-read data to assemble highly accurate and exceedingly long contiguous sequences [20]. In fungi, this approach has allowed for the assembly of chromosome-scale sequences, often spanning from telomere to telomere [22,23,24,25]. Such complete or near-complete assemblies thus allow for investigations into the structural and architectural properties of genomes, and the role these play in the biology of fungi, as well as their overall genome evolution [22,26].

Insights into the genome architecture and sub-genomic compartmentalization of *F. circinatum* have been largely limited by not having access to high-quality chromosome-level assemblies. Although genomes produced using long-read systems are available for a number of *F. circinatum* strains, these were only ordered into pseudomolecules using those produced for *F. circinatum* FSP34 [27]. Therefore, the overall goal of this study was to provide complete or near-complete, fully annotated, chromosome-level genome assemblies for *F. circinatum* by focusing specifically on the two strains (FSP34 and KS17) most frequently used in laboratory and computational studies. Accordingly, our study had three specific objectives: (i) to assemble the genomes of strains FSP34 and KS17 by making use of both second- and third-generation sequencing technologies; (ii) to annotate the genomes in terms of gene content, chromosome number and chromosome identity, as well as their telomeric and centromeric features; and (iii) to compile for both strains a detailed set of relevant biological and ecological metadata (i.e., a description of the attributes of the samples from which the genome data were generated). The latter is increasingly regarded as essential for achieving timely and impactful outcomes in genome-based investigations [28,29]. This study would thus aid future research into the chromosomal architecture, as well as the genic content, of each genome, delivering essential resources for studying this destructive plant pathogen.

## 2. Materials and Methods

### 2.1. Genome Sequencing and Assembly

DNA was extracted from *F. circinatum* strains FSP34 and KS17 as described previously [27]. These DNAs were then subjected to ONT MinION sequencing, as well as the Illumina HiSeq 2500 sequencing of a 550 bp paired-end library (Macrogen, Seoul, Republic of Korea). The data produced by the two systems were then used in a two-step process to compile the respective genomes into assemblies. First, the ONT MinION reads were trimmed and assembled using Canu v1.7.1 [30], with the “correctedErrorRate” setting adjusted, until assemblies were obtained that displayed the expected macrosynteny known in the FFSC [10,18]. The latter was determined using the LASTZ v1.02.00 plugin [31] of Geneious v7.1.9 [32]. The second step utilized the quality-filtered Illumina reads (>18 bp) returned by CLC Genomics Workbench v8.0.1 (CLCBio, Aarhus, Denmark). These reads were indexed and aligned to the Canu assembly using BWA [33] and SAMtools [34]. We then used Pilon v1.22 [35] to correct for the occurrence of sequencing errors inherent to the MinION sequencing platform [21]. Where needed, further scaffolding was performed using LASTZ-based alignments and MUMmer v4.x [36] to order and orient contigs into pseudomolecules and to pinpoint assembly breaks. The latter were indicated in the eventual assemblies with the insertion of 100 ambiguous nucleotides (i.e., 100 Ns).

MUMmer was used to evaluate the occurrence and position of a specific chromosomal translocation known to characterize the genomes of *Fusarium* species in the American clade of the FFSC [18]. This clade represents one of the so-called biogeographic clades that mainly comprises species isolated from plant hosts originating from North and South America [37]. This current study only utilized those American clade species for which high-quality genome assemblies were available (i.e., *Fusarium pilosicola, Fusarium marasasianum, Fusarium pininemorale, Fusarium sororula, Fusarium fracticaudum* and *F. temperatum*) (Appendix A). Chromosomal translocation was also investigated in a number of other *F. circinatum* strains (FFRA, FSOR, UG10, UG27, CMWF560, CMWF567, CMWF1803 and GL1327) for which suitable genome data were available (Appendix A).

### 2.2. Evaluation of Genome Quality and Completeness

Assembly completeness was estimated using the Benchmarking Universal Single-Copy Orthologs (BUSCO) v3 tool, with the “Sordariomyceta” dataset [38]. We also investigated the level to which improved sequencing technologies allowed for the enhancement of the FSP34 and KS17 assemblies. The analyses included the previous [16] and current assemblies for FSP34 (denoted as FSP34_previous_ and FSP34_current_), and the previous and current assemblies for KS17 (denoted as KS17_previous_ and KS17_current_).

### 2.3. Genome Annotation

We identified the putative positions of telomeres and centromeres for each of the pseudomolecules. Telomeres were identified by the occurrence of telomeric repeats (TTAGGG)*_n_* commonly found in filamentous fungi [39] using a sliding window of 1000 bp with 500 bp increments. To control for the spurious appearance of this repeat in the genome, telomeric caps were identified as those stretches of DNA containing a higher frequency of the telomeric repeats (i.e., ≥three times the average telomeric density across each pseudomolecule). The putative positions of centromeres were determined as described previously [40]. Briefly, this involved identifying regions with reduced G+C content and increased frequencies of mutations resembling those caused by repeat-induced point (RIP) mutations. We then determined whether the identified regions occurred at comparable locations to those of the known centromeres of *F. fujikuroi* and *F. verticillioides* [10]. The latter was achieved by using tBLASTn searches (implemented in CLC Genomics Workbench; E < 1 × 10^−5^) to compare the positions of genes flanking the centromeres of *F. fujikuroi* and *F. verticillioides* [10] with those flanking the putative centromeric regions of FSP34 and KS17. The combination of G+C content depletion, increase in RIP mutations and substantiating centromeric positions using the presence of centromeric flanking genes of structurally annotated centromeres of *F. fujikuroi* and *F. verticillioides*, confirmed that these were the centromeric regions for *F. circinatum* FSP34 and KS17. Nomenclature for centromeric positions followed those originally published in Hereditas [41].

The genes encoded by the FSP34 and KS17 genomes were identified using MAKER v2.31.8 [42]. This pipeline incorporated the gene prediction programs AUGUSTUS v3.2.2 [43], GeneMark ES [44] and SNAP [45]. In addition, predicted protein evidence from *F. graminearum* and *F. verticillioides* [4], *F. fujikuroi* [10], *F. mangiferae* and *F. proliferatum* [12], as well as *F. circinatum* [15], was utilized. The predicted genes were then functionally annotated using Blast2GO [46]. Where relevant, gene ontology (GO) term enrichment was evaluated using two-sided Fisher’s exact tests in Blast2GO (*p* < 0.05), adjusted for multiple sample testing using the Benjamini–Hochberg False Discovery Rate (FDR) analysis, and summarized with the REVIGO web server [47]. Additionally, the repeat content of the two assemblies were analyzed using the REPET v2.5 pipeline [48,49]. This pipeline allowed for both the detection and annotation of the repeats and other transposable elements (TEs).

### 2.4. Compilation of Ecological and Biological Metadata for FSP34 and KS17

In order to complement the high-quality genomes generated in this study, all relevant source and biological data for the two strains were collected from the literature. For this purpose, we focused on their origins in terms of geography, host species and tissue type, as well as the population genetics of their source populations. We also collected information generated using previous laboratory-based experimentation regarding their reproductive biology.

In this current study, we additionally investigated the pathogenicity and growth rate of these two strains. For this purpose, pathogenicity tests using six-month-old *Pinus patula* were conducted as previously described [50]. The mycelial growth of the two strains were evaluated at a range of temperatures (10 °C to 35 °C, at 5 °C intervals) on a potato dextrose agar medium (20% *w*/*v* PDA [Biolab] and 5% *w*/*v* agar [BD Difco]). For this purpose, a 6 mm mycelial plug, taken from the actively growing margin of a 7-day-old PDA culture, was placed in the center of a 90 mm Petri plate containing PDA and incubated at the respective temperatures for 7 d in the dark. Replicates of five plates per strain at each temperature was performed. Colony diameters were recorded as the average of two measurements taken along two axes at right angles to each other.

## 3. Results

### 3.1. Chromosome-Level Assemblies for FSP34 and KS17

This Whole Genome Shotgun project has been deposited at DDBJ/ENA/GenBank under the accession AYJV00000000 and LQBB00000000 for *F. circinatum* FSP34 and KS17, respectively. The respective assemblies were 45,020,843 and 44,380,849 nucleotides in length (Table 1). For FSP34, this represented an increase in genome size, with the FSP34_previous_ being 1.02% smaller than the one reported here (i.e., FSP34_current_). The opposite was observed for KS17, with the highly fragmented KS17_previous_ assembly being 1.04% larger than the KS17_current_ assembly. In terms of G+C content, a substantial difference was also observed between the old and new KS17 assemblies, but not for FSP34. G+C content values of 47.41%, 47% and 47.26% were obtained for the FSP34_previous_, FSP34_current_ and KS17_current_ assemblies, respectively, while the KS17_previous_ assembly had a G+C content of 44.69%.

Both of the new assemblies displayed N50 values exceeding 4.3 Mb (i.e., contigs ≥ 4.3 Mb in length cover at least 50% of the assembly). This high contiguity in the two assemblies gave rise to FSP34_current_ and KS17_current_ assemblies containing substantially fewer scaffolds (49 and 96 scaffolds, respectively) than the previous assemblies (Table 1). When using these scaffolds to compile each of the twelve known chromosomes for *F. circinatum* (Table 2), the pseudomolecules for the FSP34_current_ assembly contained twelve-fold fewer scaffolds than was the case for the FSP34_previous_ assembly.

In contrast to the FSP34_previous_ assembly, only a small proportion of contigs (representing less than 0.60% of the total genome size) could not be incorporated into the pseudomolecules. Also, for FSP34_current_, the pseudomolecules were larger in size than was predicted from FSP34_previous_, with pseudomolecule 12 the only exception (i.e., it was slightly smaller in the FSP34_current_ assembly). The sizes of the pseudomolecules of the FSP34_current_ and KS17_current_ assemblies had the same general trend, although pseudomolecule 12 of the KS17_current_ assembly was almost 1.66 times larger due to the presence of distal and proximal regions that were not found in its FSP34 counterpart (Figure 1).

Overall, the FSP34_current_ and KS17_current_ assemblies were highly syntenic, as indicated by the MUMmer analysis (Figure 1). Also, the reciprocal translocation between chromosomes 8 and 11 [18], previously suggested to occur in FSP34_previous_ [16], was found in the new assemblies. Both pseudomolecule 8 and 11 have scaffolds traversing the breakpoint in FSP34_current_, and the same was true for these pseudomolecules in KS17_current_.

### 3.2. Identification of Telomeres and Centromeres

In order to determine whether the 12 pseudomolecules for each assembly had their telomeric caps, we analyzed the distribution of the eukaryotic telomere repeat motif (Figure 2). The data indicated that several of the pseudomolecules compiled for the two assemblies represented chromosomes that were sequenced end-to-end. A total of 17 and 18 (out of the expected 24) telomeric caps were identified in the FSP34_current_ and KS17_current_ assemblies, respectively. These were present at both ends of pseudomolecules 7, 10, 11 and 12 of both assemblies, as well as for FSP34′s pseudomolecules 4 and 9, and KS17′s pseudomolecules 1, 3 and 6. This was a substantial improvement over the FSP34_previous_ assembly for which only seven of the twenty-four ends had telomere caps, while in the KS17_previous_ assembly, a telomere annotation was not performed due to its highly fragmented nature.

Based on G+C content depletion, combined with comparisons to the structurally annotated chromosomes of closely related species, we predicted the putative locations of the centromeric regions in all twelve of the pseudomolecules in each of the two genome assemblies (Figure 2; Appendix A). Analyses of regions corresponding to centromeres showed that these regions were less syntenic, gene-poor, G+C-depleted and were richer in repeats and RIP mutations that the rest of the respective pseudomolecules. Specifically, all of the centromeric regions in the two genome assemblies had a G+C content below 20% (Appendix A) compared to the 47% average for the genome-wide G+C content (Table 1). Also, these regions showed a high level of conservation in terms of the genes flanking them when compared to those neighboring the centromeres of *F. fujikuroi* and *F. verticillioides* (Appendix A). The only exceptions were for pseudomolecules 8, 11 and 12, where the observed synteny and collinearity differed substantially from what is known for FFSC species (see below).

Most of the predicted centromeric regions in the assemblies were located in intermediate locations on the pseudomolecules (i.e., submetacentric, metacentric and subtelomeric in pseudomolecules 1, 2 and 3, respectively) [41]. In most instances, similar positions for centromeres were also predicted for corresponding pseudomolecules in the two assemblies, except for pseudomolecules 5, 8, 11 and 12 (Appendix A). Pseudomolecule 5 of FSP34_current_ had a metacentric position, but a submetacentric position in KS17_current_. This was supported by the fact that the conserved gene flanking the proximal side of the centromere was not detected in the KS17_current_ assembly (Appendix A), implying that the shift in centromere location was associated with altered synteny on pseudomolecule 5.

In pseudomolecule 12 of FSP34, the centromere is located distally with a telocentric position (centromere present at the end of a chromosome), while it is acrocentric in KS17. In both of the pseudomolecule 12 assemblies, the corresponding centromeric gene from the distal side of *F. fujikuroi*/*F. verticillioides* chromosome 12 (Appendix A) is not present. This genomic region also corresponds to the area where the size variation was observed for this chromosome in *F. circinatum* FSP34 and KS17 (Figure 1, Table 2), suggesting that the loss of the distal portion of the chromosome in FSP34 occurred in close proximity to the centromere (see below).

In pseudomolecule 8, the centromeric region is predicted to be telocentric in the FSP34_current_ assembly, compared to being submetacentric in the KS17_current_ assembly (Appendix A). In terms of pseudomolecule 11, the predicted centromeric regions were submetacentric for FSP34_current_ and metacentric in KS17_current_. Conserved genes flanking the distal sides of the centromere for pseudomolecule 8, as well as the proximal sides of pseudomolecule 11, were not syntenous. These findings supported the presence of the reciprocal translocation between these chromosomes, thus showing synteny to centromeric genes from chromosomes 11 and 8, respectively. The reciprocal translocation in chromosome 11 was uniform for *F. circinatum* FSP34 and KS17, but the centromeric region of chromosome 8 had an internal position in KS17 (see arrangement A in Figure 3). In FSP34, it was located at the distal end of the chromosome in a telocentric position (arrangement B in Figure 3).

A comparison of a diverse set of *F. circinatum* strains, as well as closely related species from the American clade FFSC species (Appendix A), revealed the presence of the reciprocal translocation between chromosomes 8 and 11 in all of the genomes examined (Figure 3). It had a conserved position in chromosome 11, but the position in chromosome 8 was variable, regardless of species or strain identity. For example, *F. circinatum* displayed three iterations of the arrangement observed for the translocation on chromosome 8 (arrangements A and B described above, and arrangement C in Figure 3). The third iteration was similar to the one observed in KS17, but the translocated portion of chromosome 11 present on chromosome 8 was inverted (Figure 3C). In addition, one of the arrangements of this translocation was only observed in *F. pininemorale*, with an additional chromosomal rearrangement observed on chromosome 8 (Figure 3D). In all these arrangements, the breakage for this reciprocal translocation appeared to be associated with the centromeres.

### 3.3. Genome Completeness and Gene Content

The genome assemblies generated here were highly complete (i.e., 97.3% for FSP34_current_ and 98.1% for KS17_current_), containing > 3624 of the set of 3725 BUSCO genes used to estimate genome completeness in the class Sordariomycetes (Table 1). This was a substantial improvement over the older genome assemblies for these fungi, particularly for the strain KS17. With a completeness score of 76.2%, the KS17_pevious_ assembly lacked almost 900 of the expected BUSCO genes.

As expected, substantially more genes were identified and annotated by MAKER in the FSP34_current_ assembly than the previous version (Table 1). The new assembly contained >500 more genes, at a slightly higher density of 344.06 genes/Mb (compared to 339.68 genes/Mb in the FSP34_previous_ assembly). Direct comparisons between the new and the older versions of the KS17 assembly could not be made, as the annotations for the older assembly utilized software lacking the same capabilities as MAKER [17]. Nevertheless, similar gene densities were estimated for the FSP34_current_ and KS17_current_ assemblies, and this similarity was also generally extended to individual sets of pseudomolecules (results not shown). Overall, pseudomolecule 10 had the highest gene density (i.e., 377.20 genes/Mb for FSP34_current_ and 382.52 genes/Mb for KS17_current_) while it was lowest on pseudomolecule 12 (i.e., 325.67 genes/Mb for FSP34_current_ and 294.02 genes/Mb KS17_current_).

Of the genes predicted for the FSP34_current_ assembly, only 642 (4.14%) lacked relevant BLAST hits to any known protein (Appendix A). In total, 4694 (30.30%) and 1204 (7.77%) genes could not be assigned GO or InterPro identifications, respectively. A similar pattern was observed for the KS17_current_ assembly (Appendix A), with 626 (4.14%) lacking BLAST hits, 5072 (33.56%) not assigned GO identifications, and 1201 (7.95%) not assigned any InterPro identifications. See Appendix A for the full lists of functional annotations for the genes predicted in the FSP34_current_ and KS17_current_ assemblies.

Due to the differences observed between pseudomolecule 12 and the remainder of the pseudomolecules in the new assemblies, we compared the gene content using two-sided Fisher’s exact tests on the GO terms. For FSP34, pseudomolecule 12 was significantly (*p* < 0.05) enriched for GO terms involved in biological functions associated with cellular aromatic compound metabolism, primary metabolism and organic cyclic compound metabolism, as well as GO terms associated with nitrogen compound metabolism, transport and localization (Appendix A; Appendix A). In the case of KS17_current_, pseudomolecule 12 was also enriched for GO terms involved in cellular amide metabolism, organic substance transport, organelle organization and protein metabolism, as well as GO terms associated with the cellular response to stimulus, catabolism and cellular catabolism (Appendix A; Appendix A).

### 3.4. Genome Repetitiveness and TE Content

Improvements in the genome assemblies were evident in their increased repeat and TE content (Table 1 and Appendix A). FSP34_previous_ had 2.81% of the genome that was characterized as repetitive elements, whilst this increased to 8.75% in FSP34_current_. The KS17_current_ assembly contained a comparable 8.60% of repetitive elements. When considering only pseudomolecules 1–11 (i.e., those representing core chromosomes), the repetitive content was 4.68–15.15% across the FSP34_current_ core set and 4.80–11.97% for the KS17_current_ core set, with the two assemblies sharing similar content patterns. Although the repeat content of pseudomolecule 12 in the FSP34_current_ assembly was 6.92%, its counterpart in the KS17_current_ assembly was 26.6% (Table 3).

The large differences in repeat content between pseudomolecule 12 in the two assemblies were examined in more detail. This revealed that the additional regions located at the proximal and distal parts of pseudomolecule 12 of KS17_current_ (see Figure 1B) accounted for most of the observed increased repetitive content (Table 3). Interestingly, the distal part of the pseudomolecule was also highly depleted in G+C content (31.88%) compared to the average of 41.73% for the entire pseudomolecule. The additional regions at the proximal and distal parts of pseudomolecule 12 of KS17_current_ were also extremely gene poor (Table 3).

Both Class I TEs (i.e., retrotransposons) and Class II TEs (i.e., DNA transposons) were identified in the FSP34_previous_ (Appendix A), FSP34_current_ (Appendix A) and KS17_current_ assemblies (Appendix A). FSP34_current_ had Class I TEs, namely TRIM (Terminal Repeat transposons in Miniature), LARD (Large Retrotransposon derivatives), LINE (Long Interspersed Nuclear Elements) and unclassified retrotransposons (non-autonomous retrotransposon), as well as Class II TE TIR (Terminal Inverted Repeats) (Figure 4A and Appendix A). KS17_current_ had LARD, unclassified retrotransposons and TIR transposable elements in very similar densities on each pseudomolecule (Figure 4B and Appendix A). In addition, the frequency of the TEs in FSP34_current_ was LARD > unclassified retrotransposons > TIR > LINE > TRIM, whilst in KS17_current_ it was LARD > TIR > unclassified retrotransposons (Appendix A).

### 3.5. Detailed Source and Biological Information for KS17 and FSP34

For summarizing available information regarding the ecology and biology of *F. circinatum* strains KS17 and FSP34, we utilized the published literature as well as the findings of our in vitro growth studies and pathogenicity tests. Details regarding their origins, reproductive biology and the population dynamics of their source populations are provided in Table 4. It should be taken into account that most of the information regarding the origins of FSP34 comes from a study by Gordon et al. [52], although they never explicitly mentioned the strain. Also, strain KS17 was isolated during a study by Steenkamp et al. [50], but it did not form part of their experiments.

Additionally, the results of our pathogenicity tests showed that both KS17 and FSP34 are pathogenic to *P. patula* seedlings (Appendix A). Three weeks after inoculation, lesions were observed for the FSP34 and KS17 treatments, respectively, while no lesions developed in the control treatment. FSP34 was, however, significantly more virulent or aggressive, with lesion lengths averaging 15.45 mm compared to the 6.2 mm of KS17 (*p* < 0.001). Likewise, compared to KS17, FSP34 grew significantly faster at all of the temperatures tested, although neither grew at 35° (Appendix A).

## 4. Discussion

By harnessing the power of second- and third-generation sequencing technologies [55,56], we determined the whole genome sequences for two important strains of *F. circinatum* (FSP34 and KS17) [15,16,17]. For both, we obtained near complete chromosome-level assemblies, consisting of contiguous sequence mostly spanning entire molecules from telomere to telomere. The two new assemblies emphasized the macrosyntenic nature of the FFSC genomes [10,18] and allowed for the detailed examination of the previously suggested reciprocal translocation between chromosomes 8 and 11 in species from the American clade of FFSC [16,18]. The data presented here further revealed significant variability in chromosome 12 of *F. circinatum*.

A comparison of the two *F. circinatum* strains revealed that their genomes were highly syntenic. This was expected for members of the same species, and it also echoed the high level of conservation of large-scale chromosomal arrangements typically observed across the FFSC, e.g., between *F. circinatum* and more distantly related FFSC species such as *Fusarium verticillioides* (causal agent of maize ear rot) and *Fusarium fujikuroi* (causal agent of bakanae disease in rice seedlings) [10,18]. Our results are also consistent with those of a study exploring intraspecific genome plasticity in *F. circinatum*, which reported that >90% of a strain’s genome sequence is conserved and “alignable” with that of another strain of the fungus [27]. Nonetheless, structural changes to several chromosomes of *F. circinatum* were evident, with most variation located on the subtelomeric regions of chromosomes 1–11, with chromosome 12 being the most variable.

The findings presented here improved our understanding of the origin and evolution of the reciprocal translocation between chromosomes 8 and 11 of species from the American clade of the FFSC. Our high-quality, chromosome-level assemblies confirmed its existence in the two *F. circinatum* strains examined. Although the translocation was mentioned in previous studies [16,18], all of the genome assemblies in question were generated using short-read sequences. However, our use of ONT MinION long-read sequences allowed for the assembly of contiguous sequences across the predicted chromosomal breakpoints. Apart from providing the strongest evidence yet for the existence of this translocation, our data also showed that the region associated with it is highly variable and that this variability is likely associated with centromeres. This was well illustrated by the different arrangements obtained for the translocated region on chromosome 8 relative to the centromeric region (see arrangements A and B in Figure 3). This centromere-associated “evolvability” of chromosomes has also been reported in other eukaryotes [57], including fungi [58]. Double-stranded breaks occurring in the repeat-rich centromeric DNA of *Cryptococcus*, for example, have been shown to mediate chromosomal translocations and rearrangements, which are thought to have played a significant role in the evolution of these fungi [59,60]. In the case of *F. circinatum* and the remainder of the American clade of the FFSC, the emergence of such as centromere-mediated translocation involving chromosomes 8 and 11 would have predated the divergence of extant species and likely occurred in the ancestor of the clade. The fact that the region varies among strains of *F. circinatum* and among other American clade species, irrespective of their species identities, suggests a continued role for centromeres in shaping the genome architecture of these fungi.

We now have a more comprehensive understanding of the repetitive nature of the *F. circinatum* genome; this is important as these repetitive elements are known to bring about changes and variations in genome architecture [61]. Indeed, the dynamic and plastic nature of fungal genomes [62] is epitomized in chromosome 12 of *F. circinatum*. This chromosome in KS17 had substantially more TEs compared to FSP34 and the presence of repetitive elements likely enabled the formation of the chromosome length polymorphism observed. In particular, this chromosome in KS17 has more Class I TEs than in FSP34, and the activity of retrotransposons is known to give rise to sequence polymorphism and genome expansion [63].

Apart from its repetitive nature, chromosome 12 of *F. circinatum* also displayed unusual length polymorphism relative to the core chromosomes. Strain KS17′s twelfth chromosome was >1.6× longer than that of FSP34. Similar patterns have been found in other members of the FFSC for chromosome 12, especially *F. fracticaudum,* where this molecule is over 1 Mb in length. In this current study, the length polymorphism was found to be associated with the proximal and distal portions of the KS17 chromosome 12. This was also evident from the genomic alignments of chromosome 12 from *F. circinatum* CMWF1803 (isolated from diseased *P. patula* branches in Hidalgo, Mexico) [64] to those of FSP34 and KS17 (see Appendix A for details). Interestingly, the “middle” portion of the KS17 chromosome 12 (i.e., the region that is syntenic to the entire chromosome 12 of FSP34), is characterized by gene density and G+C content, and repeats content that is similar to those of chromosomes 1–11. Therefore, FSP34 chromosome 12 seems to have either lost the regions enabling variation, or could represent an ancestral or fundamental version of chromosome 12. A possible role for centromeric elements in these chromosome length polymorphisms cannot be excluded, as chromosome 12 of FSP34 has a centromere at one of its ends and not a telomere, as in KS17.

The biological functions associated with chromosome 12 of *F. circinatum* remain unclear. In the FFSC, the twelfth chromosome is generally regarded both as dispensable and strain-specific [65,66]. In notable pathogens such as *F. oxysporum*, such chromosomes have been shown to be involved in pathogenicity, in part due to the small effector proteins they encode [4]. In chromosome 12 of *F. circinatum* FSP34 and KS17, there was an increase in genes involved in different processes and functions between them, with no clear-cut enrichment of genes specific for chromosome 12. However, the elimination of chromosome 12 from the genome of the fungus has been linked to the reduced virulence on *P. radiata* seedlings [67]. Clearly, further research is needed in order to clarify the evolution and role of this chromosome in *F. circinatum*.

The improved *F. circinatum* assemblies provided in this study represent an invaluable resource for genomic comparisons in the FFSC. We have shown the benefit of resequencing previous genomic sequence data generated using second-generation sequencing technologies, and highlighted various structural chromosomal differences shown using the ONT MinION sequencing technology, in conjunction with Illumina HiSeq for error correction. This was especially true for KS17, which was formerly highly fragmented and incomplete. Studies on intraspecific genetic variation has the capacity to provide information on how genetic variation is accumulated within genomes, and to address how this impacts the adaptability and variability of fungi [27,68]. Combined with the detailed metadata compiled for the two strains (see Table 4), our high-quality chromosome-level assembly for *F. circinatum* will undoubtedly facilitate more comprehensive genomic and comparative studies into this destructive plant pathogen.

## Figures and Tables

**Figure 1 pathogens-13-00070-f001:**
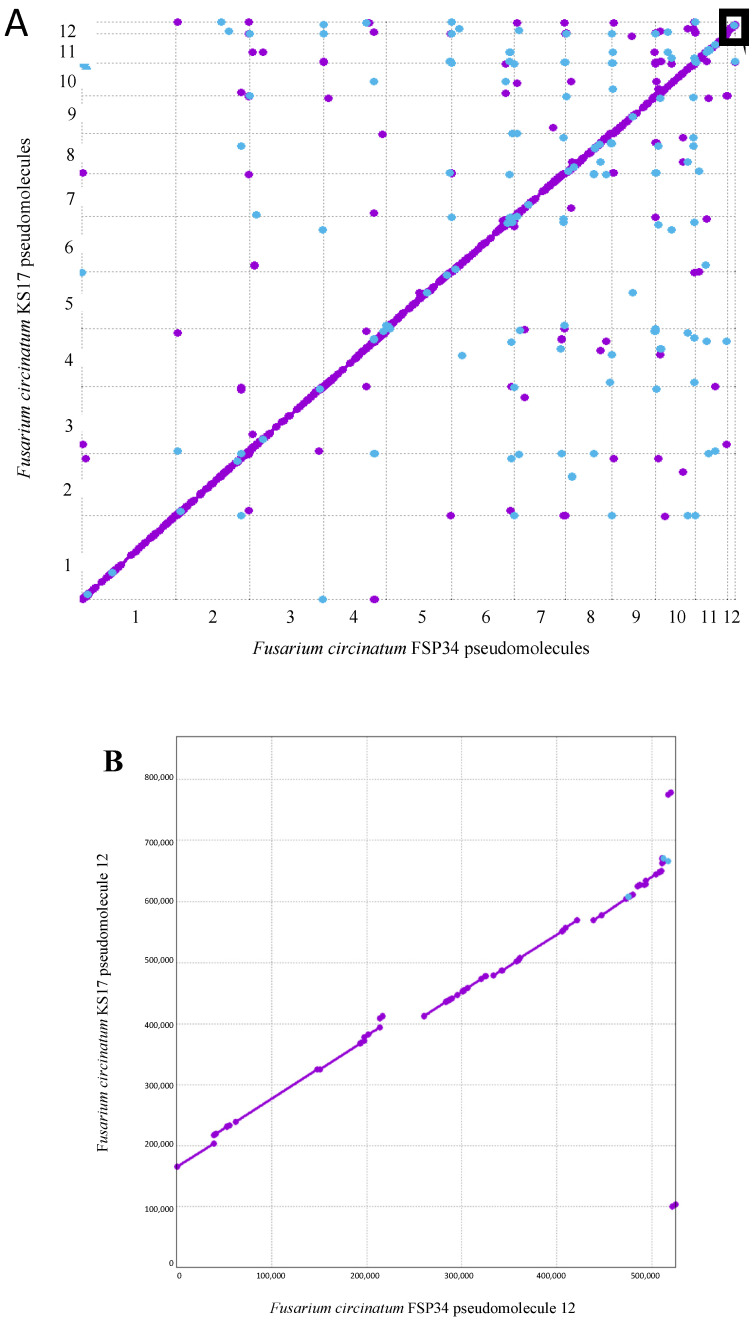
(**A**) Sequence comparison between the set of 12 pseudomolecules compiled for each genome. MUMmer revealed high levels of synteny across the *F. circinatum* FSP34 and KS17 assemblies. Forward matches are indicated with purple dots and reverse matches with blue dots. A close up of the black box is shown in (**B**). (**B**) Close-up of the comparison between the twelfth pseudomolecule from the two genome assemblies.

**Figure 2 pathogens-13-00070-f002:**
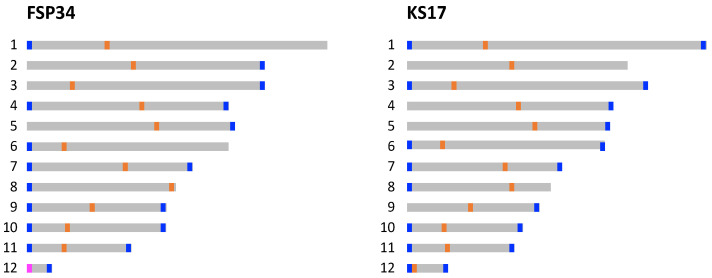
Schematic overview of the relative positions of centromeres (blue bars) and telomeres (orange bars) identified and/or predicted for each of the 12 pseudomolecules in the FSP34 and KS17 genome assemblies generated in this study (pink bar in FSP34 pseudomolecule 12 is indicative that the centromere and telomere are both positioned distally in close proximity to each other).

**Figure 3 pathogens-13-00070-f003:**
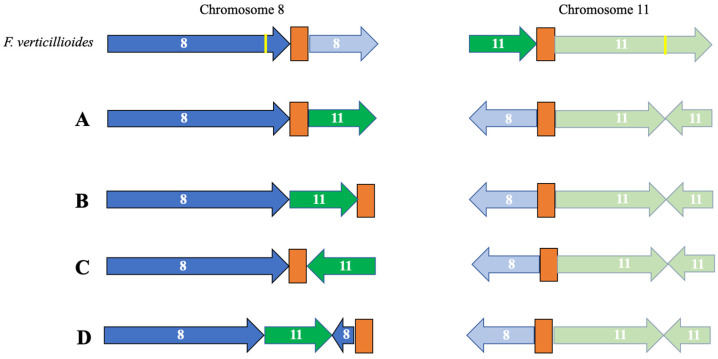
Schematic representation of the reciprocal translocation between chromosomes 8 and 11 in various American clade species of the FFSC (Appendix A) relative to *F. verticillioides*. Portions of chromosomes originating from chromosome 8 in *F. verticillioides* are indicated in blue, whilst those originating from chromosome 11 in *F. verticillioides* are indicated in green. The centromeric regions are indicated in orange. The yellow line represents chromosomal breakpoints not involving centromeric regions. Only chromosomal arrangements larger than 200,000 bp are indicated in this representation. Arrangement A represents the translocation in *F. circinatum* strains KS17, FFRA, UG10, UG27, CMWF560, CMWF1803 and GL1327, as well as *F. pilosicola*, *F. temperatum*, *F. marasasianum* and *F. sororula*. Arrangement B represents the translocation in *F. circinatum* FSP34. Arrangement C represents the translocation in *F. circinatum* strains FSOR and CMWF567, as well as *F. fracticaudum*, while arrangement D represents the one in *F. pininemorale*.

**Figure 4 pathogens-13-00070-f004:**
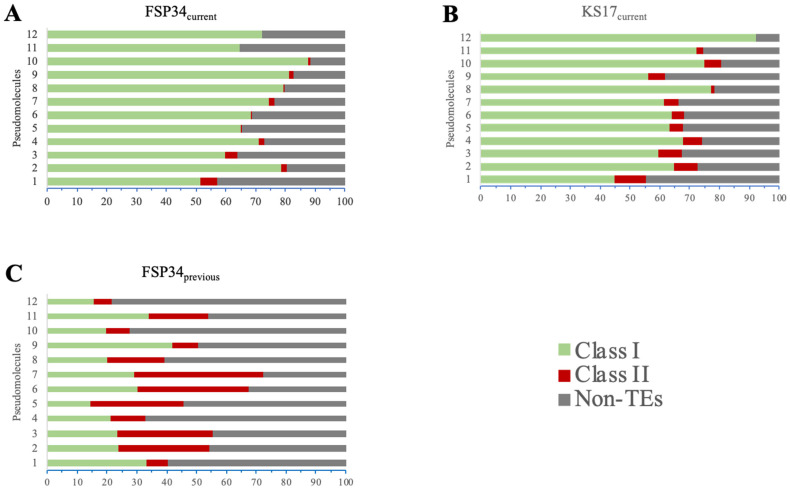
Classes and orders of transposable elements (TEs) identified in FSP34_current_ (**A**), KS17_current_ (**B**) and FSP34_previous_ (**C**), based on the classification of Wicker et al. [51]. The TEs are given as Class I (TRIM—Terminal Repeat transposons in Miniature, LARD—Large Retrotransposon derivatives, LINE—Long Interspersed Nuclear Elements and unclassified—non-autonomous retrotransposon), and Class II (TIR—Terminal Inverted Repeats).

**Table 1 pathogens-13-00070-t001:** Genome statistics of *F. circinatum* strains FSP34 and KS17.

Genome Statistic	FSP34		KS17	
	Previous ^1^	Current	Previous ^2^	Current
Genome size (bp)	43,932,912	45,020,843	46,325,048	44,380,849
G+C content (%)	47.41	47.00	44.69	47.26
Number of open reading frames	14,923 ^3^	15,490	16,502 ^4^	15,113
Gene density (orfs/Mb)	339.68	344.06	356.22	340.53
Number of scaffolds	585	49	6033	96
N50 (bp)	363,633	4,313,168	95,695	4,401,926
Average scaffold size (bp)	75,085	1,667,439	7679	1,431,640
Unmapped scaffolds (total % of genome)	418 (3.03%)	15 (0.60%)	- ^5^	19 (0.78%)
Repeat content ^6^	2.81%	8.75%	-	8.60%
Genome completeness ^7^	94.8%	97.3%	76.2%	98.1%

^1^ Wingfield et al. [16]. ^2^ Van Wyk et al. [17]. ^3^ Annotated using MAKER [15]. ^4^ Annotated using WebAUGUSTUS (https://bioinf.uni-greifswald.de/webaugustus/prediction/create, accessed on 1 November 2017) previously [17]. ^5^ Scaffolds were never assigned to chromosomes [17]. ^6^ Determined using REPET v2.5 [48,49]. Assembly of KS17_previous_ was too fragmented to determine the repeat content. ^7^ Based on BUSCO v3.0.2 using the “Sordariomycete” database [38].

**Table 2 pathogens-13-00070-t002:** Size (base pairs) of the pseudomolecules from *F. circinatum* strains FSP34 and KS17.

Pseudomolecule	FSP34 ^1^		KS17_current_
	Previous	Current	
1	6,190,704 (14)	6,407,689 (2)	6,397,914 (8)
2	4,773,114 (22)	5,066,197 (3)	4,709,326 (5)
3	4,756,822 (18)	5,081,888 (3)	5,148,568 (3)
4	4,140,424 (12)	4,313,168 (2)	4,401,926 (5)
5	4,399,406 (10)	4,432,553 (2)	4,304,443 (7)
6	4,053,349 (21)	4,301,895 (3)	4,219,930 (9)
7	3,472,423 (19)	3,541,054 (5)	3,312,103 (8)
8	3,024,507 (17)	3,172,915 (5)	3,066,990 (9)
9	2,773,158 (8)	2,981,544 (1)	2,828,005 (4)
10	2,371,510 (16)	2,698,820 (3)	2,483,521 (7)
11	2,122,486 (11)	2,228,420 (3)	2,291,537 (5)
12	525,791 (6)	525,065 (1)	870,680 (3)

^1^ The previous FSP34 genome was sequenced by Wingfield et al. [16]. Number of scaffolds used for building pseudomolecules are indicated in brackets.

**Table 3 pathogens-13-00070-t003:** G+C and repeat content, and gene density of pseudomolecule 12 of FSP34 and KS17.

Content Estimates	FSP34 Pseudomolecule 12 ^1^	KS17 Pseudomolecule 12 ^2^			
		Whole Molecule	Distal	Middle	Proximal
G+C content (%)	46.36	41.73	31.88	45.18	41.48
Gene density (orfs/Mb)	325.67	294.01	123.53	349.52	300.00
Repeats (%)	6.92	26.63	59.67	12.67	33.48

^1^ FSP34_previous_ has pseudomolecule 12 assembled into one scaffold with a telomeric cap at both ends. ^2^ The “distal” portion of KS17_current_ pseudomolecule 12 corresponds to the first 170,000 bp that has a length polymorphism compared to FSP34_current_ pseudomolecule 12. “Middle” is the portion of KS17_current_ pseudomolecule 12 corresponding to position 170,001–670,679 bp displaying synteny to the FSP34_current_ pseudomolecule 12. “Proximal” in KS17_current_ pseudomolecule 12 corresponds to the last 200,000 bp, which has a length polymorphism in comparison to FSP34_current_ pseudomolecule 12.

**Table 4 pathogens-13-00070-t004:** Summary of the available metadata for *F. circinatum* strains FSP34 and KS17.

Data/Properties	FSP34	KS17 ^1^	References
Strain origin			
Collection date	Unknown date between March 1993 and April 1995	October 2005	[50,52]
Collector	TR Gordon	OM Mashandula, ET Steenkamp	[52]
Host plant	*Pinus radiata*	*Pinus radiata*	[52]
Host tissue	Tissue from the leading edge of a canker on the branch of a mature tree	Diseased root tissue of a nursery seedling	[52]
Geographic location	Monterey, California (USA)	Karatara, Western Cape (South Africa)	[52]
Location description	Exact location is unknown, but collected in a region where mature trees of native *Pinus* species displayed symptoms of pitch canker	Commercial seedling nursery; collected during an outbreak of *F. circinatum*-associated root disease	[52]
Reproductive biology			
Mating type	*MAT1-1*	*MAT1-2*	[15] (unpublished)
Fertility	Male fertile; mostly female sterile; also capable of mating with sexually compatible strains of *Fusarium temperatum* to produce fertile hybrid progeny.	Male fertile; displays some level of fertility as a female.	[53] (unpublished)
Source population dynamics	Forms part of a population with limited genetic diversity that propagates asexually	Forms part of a moderately diverse population that propagates mainly asexually.	[54]
Growth in culture	Grows at 5–30 °C, but not at 35 °C; grows faster than KS17 at 15–30 °C.	Grows at 5–30 °C, but not at 35 °C; grows slower than FSP34 at 15–30 °C.	This study
Pathogenicity	Capable of inducing lesions when inoculated onto the apices of the main stems of *P. patula* seedlings; more virulent than KS17.	Capable of inducing lesions when inoculated onto the apices of the main stems of *P. patula* seedlings; less virulent than FSP34.	This study

^1^ *Fusarium circinatum* strain KS17 was isolated at the same time and location, and from the same tissue type as those used in the study by Steenkamp et al. [50] but did not form part of that study.

## Data Availability

The Whole Genome Shotgun project for *Fusarium circinatum* has been deposited at DDBJ/ENA/GenBank under the accession AYJV00000000 and LQBB00000000 for *F. circinatum* FSP34 and KS17, respectively.

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
