# Peer review of "Chromosome-Level Assemblies for the Pine Pitch Canker Pathogen Fusarium circinatum"

_pathogens, 2024, doi:10.3390/pathogens13010070_

Round 1
Reviewer 1 Report
Comments and Suggestions for Authors
This work was focused on improving the genomic understanding of Fusarium circinatum, a pathogen known to be a significant threat to pine-based forestry. The findings have implications for understanding the pathogen's biology and could inform strategies for mitigating its impact on pine-based forestry.
Please provide the accession numbers of sequenced cultures.
The statistical analysis of gene ontology (GO) term enrichment using Fisher’s exact tests is appropriate. However, it would be helpful to mention the correction method applied for multiple testing (e.g., Bonferroni correction) to address potential false positives.
Providing a brief overview of the criteria used to predict centromeres, especially regarding G+C content depletion and comparisons to structurally annotated chromosomes, would enhance the understanding of the methodology.
The statement that direct comparisons between the new and older versions of the KS17 assembly could not be made raises questions. If possible, providing a brief explanation of the reasons for this limitation would clarify the issue
Including details on the version of the BUSCO dataset and the specific parameters used in the analysis would enhance the reproducibility of the completeness assessment.
Comments on the Quality of English Language
Minor corrections required.
Author Response
Reviewer 1
This work was focused on improving the genomic understanding of Fusarium circinatum, a pathogen known to be a significant threat to pine-based forestry. The findings have implications for understanding the pathogen's biology and could inform strategies for mitigating its impact on pine-based forestry.
Please provide the accession numbers of sequenced cultures.
RESPONSE: The accession numbers for FSP34 and KS17 were provided in the Results section (Lines 197-198). The strains FSP34 and KS17 have the accession numbers AYJV00000000 and LQBB00000000, respectively. In addition, we have contacted NCBI and requested them to publicly release of these two genomes.
The statistical analysis of gene ontology (GO) term enrichment using Fisher’s exact tests is appropriate. However, it would be helpful to mention the correction method applied for multiple testing (e.g., Bonferroni correction) to address potential false positives.
RESPONSE: This was addressed in lines 173-176 where we added the phrase “adjusted for multiple sample testing using the Benjamini-Hochberg False Discovery Rate (FDR) analysis”.
Providing a brief overview of the criteria used to predict centromeres, especially regarding G+C content depletion and comparisons to structurally annotated chromosomes, would enhance the understanding of the methodology.
RESPONSE: The criteria for centromere prediction was clarified in lines 162-166. We added the sentence: “The combination of G+C content depletion, increase in RIP mutations and substantiating centromeric positions by the presence of centromeric flanking genes of structurally annotated centromeres of F. fujikuroi and F. verticillioides, confirmed that these were the centromeric regions of F. circinatum FSP34 and KS17.”
The statement that direct comparisons between the new and older versions of the KS17 assembly could not be made raises questions. If possible, providing a brief explanation of the reasons for this limitation would clarify the issue
RESPONSE: This was clarified in lines 514-515 where we highlight that the older version of the KS17 assembly was highly fragmented and incomplete.
Including details on the version of the BUSCO dataset and the specific parameters used in the analysis would enhance the reproducibility of the completeness assessment.
RESPONSE: The relevant information was included in line 141 where state that “Assembly completeness was estimated using the Benchmarking Universal Single-Copy Orthologs (BUSCO) v3 tool, with the “Sordariomyceta” dataset [38].” Additionally, we revised the footnote of Table 1, line 217 to read as follows: “Based on BUSCO v3.0.2 using the “Sordariomycete” database [38].”
Reviewer 2 Report
Comments and Suggestions for Authors
The manuscript titled “Chromosome-level assemblies for the pine pitch canker 2 pathogen Fusarium circinatum” presented F. circinatum assemblies addressing and resolving some of the issues, mainly highly fragmented or structurally ill-defined previously available genome sequences. This study provided long read sequences as well as short reads using Oxford Nanopore Technologies MinION and Illumina.
The manuscript is well written, with adequate description of methods, well covered introduction, and discussion.
Author Response
Reviewer 2
RESPONSE: We would like to express our gratitude for the time and effort invested in reviewing our manuscript.
Reviewer 3 Report
Comments and Suggestions for Authors
Dear authors,
the manuscript with the title 'Chromosome-level assemblies for the pine pitch canker pathogen Fusarium circinatum', which you submitted, presents valuable information on using new technology to improve genome assemblies, for which the interesting case of F. circinatum is a good example. The manuscript is presenting valuable methodical insights and is well written.
Providing and updating assemblies, interpretation of data and also providing biological data and relevant literature is often not considered as important, compared to the sequence data itself, which is accessible for everyone. Therefore, I consider your work as an important contextualisation for your data and valuable for the readers. It gives the reader the opportunity to reproduce the data or use the workflow for their own research.
I found some minor errors and some figures need to be improved. Please check my comments to the pdf file and make corrections accordingly. There is also some language editing required.
Sometimes results and discussion sections are a bit mixed up (e.g. line 204-205). Please check and transfer your interpretations to the discussion section where necessary.
Kind regards,
The reviewer

Comments on the Quality of English Language
Dear authors,
I found some grammar, punctuation and hyphenation errors throughout the entire text (e.g. line 260, 263, 351, 421, 459). Please check the manuscript again for other errors that I didn’t highlight. All species names have to be written in italics (e.g. Caption of Figure 3). Furthermore, a free space has to be left between numbers and units like mm or °C, and also when you write numbers over 1000 -> 1 000. You did this right mostly but there are some exceptions.
Kind regards,
The reviewer
Author Response
Reviewer 3
The manuscript with the title 'Chromosome-level assemblies for the pine pitch canker pathogen Fusarium circinatum', which you submitted, presents valuable information on using new technology to improve genome assemblies, for which the interesting case of F. circinatum is a good example. The manuscript is presenting valuable methodical insights and is well written.
Providing and updating assemblies, interpretation of data and also providing biological data and relevant literature is often not considered as important, compared to the sequence data itself, which is accessible for everyone. Therefore, I consider your work as an important contextualisation for your data and valuable for the readers. It gives the reader the opportunity to reproduce the data or use the workflow for their own research.
I found some minor errors and some figures need to be improved. Please check my comments to the pdf file and make corrections accordingly. There is also some language editing required:
Line 15-17: Please check if affiliations can be combined. If so, authors 5,6 and 7 would get the same affiliation number, in that case 5.
RESPONSE: The suggested revision was made (Line 6-9).
Line 135: please use another formulation.
RESPONSE: To address this issue, w used the phrase "from North and South America" (lines 133-134).
Line 158: no underlines to indicate letters in an abbreviation.
RESPONSE: The suggested revision was made (lines 156-157).
Figure 1: Please reorganize this figure in order to make everything relevant also readable. X and y achsis labelling must not overlap. Overall resolution also needs to be improved.
The graph in the top right corner is not readable. Please provide this in the same way as the figure on the left according to my comments above. Either youo could provide both as panel figure A/B or alternatively one as supplementary material.
RESPONSE: To address the issue, the entire figure was revised and is now presented in two separate panels, labelled A and B.
Figure 3 legend species names are not in italics.
RESPONSE: This has been corrected.
Figure 4: Non-TEs are not visible in the graphs. If it is not relevant here, please delete it from the legend or chose another color to improve the visibility for the reader.
RESPONSE: In the graphs, Non-TEs are now darker and visible.
Line 440: It is not common to address the reader directly. Please consider rewriting using passive voice (e.g. It should be taken into account that...)
RESPONSE: The suggested revised was made (line 441).
Line 457: Fix strikethrough-additional word “strain”
RESPONSE: The suggested revised was made (Line 454).
Sometimes results and discussion sections are a bit mixed up (e.g. line 204-205). Please check and transfer your interpretations to the discussion section where necessary.
RESPONSE: Elements representing “discussion” was removed from the Results section.
I found some grammar, punctuation and hyphenation errors throughout the entire text (e.g. line 260, 263, 351, 421, 459). Please check the manuscript again for other errors that I didn’t highlight. All species names have to be written in italics (e.g. Caption of Figure 3). Furthermore, a free space has to be left between numbers and units like mm or °C, and also when you write numbers over 1000 -> 1 000. You did this right mostly but there are some exceptions.
RESPONSE: The manuscript has been checked again and corrections were made in lines 26, 30, 48, 116, 122, 123, 153, 157, 177, 188, 215, 218, 228, 258, 262, 362, 376 and 378, as well as in Table 4.
Round 2
Reviewer 1 Report
Comments and Suggestions for Authors
The manuscript has been revised significantly and can be accepted in its current form.